# Differential diffusion driven far-from-equilibrium shape-shifting of hydrogels

Yue Zhang[1,2], Kangkang Liu[1], Tao Liu[3], Chujun Ni[1], Di Chen[1], Jiamei Guo[3], Chang Liu[1], Jian Zhou[1], Zheng Jia [3], Qian Zhao [1,2 ✉], Pengju Pan[1] & Tao Xie [1,2]

Far-from-equilibrium (FFE) conditions give rise to many unusual phenomena in nature. In contrast, synthetic shape-shifting materials typically rely on monotonic evolution between equilibrium states, limiting inherently the richness of the shape-shifting behaviors. Here we report an unanticipated shape-shifting behavior for a hydrogel that can be programmed to operate FFE-like behavior. During its temperature triggered shape-shifting event, the programmed stress induces uneven water diffusion, which pushes the hydrogel off the equilibrium based natural pathway. The resulting geometric change enhances the diffusion contrast in return, creating a self-amplifying sequence that drives the system into an FFE condition. Consequently, the hydrogel exhibits counterintuitive two opposite shape-shifting events under one single stimulation, at a speed accelerated by more than one order magnitude. Our discovery points to a future direction in creating FFE conditions to access otherwise unattainable shape-shifting behaviors, with potential implications for many engineering applications including soft robotics and medical devices.

[1] State Key Laboratory of Chemical Engineering, College of Chemical and Biological Engineering, Zhejiang University, Hangzhou 310027, China. [2] ZJU-Hangzhou Global Scientific and Technological Innovation Center, Hangzhou 311215, China. [3] Key Laboratory of Soft Machines and Smart Devices of Zhejiang Province, Center for X-Mechanics, Department of Engineering Mechanics, Zhejiang University, Hangzhou 310027, China. ✉email: qianzhao@zju.edu.cn

Active materials that can undergo shape-shifting upon external triggering play increasingly important roles in modern technologies including bioelectronics[1], sensors[2], medical devices[3], and soft robotics[4–7]. Design/discovery of unanticipated shape-shifting behaviors can open up new opportunities in many engineering fields, consequently has been a constant pursuit[8–10]. Elegant molecular enabling mechanisms, along with advances in external controlling techniques, have led to materials that exhibit ever more sophisticated shape-shifting behaviors[11–13]. Amongst various shape-shifting materials, stimuli-responsive hydrogels have attracted enormous attention because of their unique water-swollen polymer networks[14–16]. In particular, the presence of water provides many opportunities to design rich stimuli-responsive behaviors. Despite such, typical hydrogels can only morph monotonically between equilibrium shapes triggered by external stimulation. One seeming exception is the introduction of a mechanical buckling mechanism in gel assemblies[17–19]. It allows converting the continuous shape evolution into an abrupt action that amplifies the mechanical output. However, the shape-shifting direction remains monotonic and the rapid buckling action requires a lengthy energy built-up process. Another notable example is self-oscillating hydrogels[20,21]. Driven by a chemical fuel, a gel can be designed to undergo self-oscillating shape-shifting actions via either an intrinsically oscillating chemical reaction or a homeostatic feedback loop[22]. Despite the uniqueness of the self-oscillation, the underlying shape-shifting still occurs monotonically between equilibrium shapes.

In contrast to the above unusual shape-shifting behaviors, we describe hereafter a far-from-equilibrium (FFE) shape-shifting behavior. Under a rather mild condition (temperature change), one single stimulation triggers two opposite non-monotonic shape-shifting events at a speed that is more than one order magnitude faster than comparative hydrogels without the mechanism.

## Results

**Characterizations of the hydrogels**. The difference between a conventional shape-shifting behavior and an FFE behavior is schematically illustrated in Fig. 1a. The former exhibits a monotonic reversible shape-shifting behavior between two equilibrium shapes corresponding to two external conditions (e.g., temperatures). The "monotonic" refers to the fact that the intermediate shapes in the kinetic shape-shifting pathway represent natural evolution between the two equilibrium shapes. For simplicity, the flower closes continuously to a final partially closed state. In stark contrast, in an FFE behavior, during shifting between the two equilibrium shapes, a third shape that is completely deviated from a continuous pathway is observed. That is, the flower changes from its original open state to an intermediate fully closed state and then to a partially closed state. In a nutshell, when triggered once, the conventional hydrogel exhibits a single close action whereas the FFE leads to close–open dual actions. As illustrated hereafter, the mechanism for the FFE behavior is rather general and the driving conditions are quite mild.

Our discovery of the surprising FFE behavior for a hydrogel is accidental. Our original intention is to develop a reversible shape-shifting hydrogel for which the pathway can be programmed. That is, the shape-shifting pathway can be altered via physical programming after the hydrogel is synthesized[23]. This type of behavior is similar to two-way shape memory polymers[24] and drastically different from conventional responsive hydrogels for which the shape-shifting pathway is determined in the material synthesis/fabrication step. Our hydrogel is a semi-interpenetrating network obtained by crosslinking

N-isopropylacrylamide (NIPAM) with N,N′-methylenebisacrylamide (MBA, the crosslinker) in the water solution of poly(vinyl alcohol) (PVA) (Fig. 1b).

Two features are noteworthy for this hydrogel: (1) poly(N-isopropylacrylamide) (PNIPAM) chains can undergo a volume phase transition (VPT); (2) PVA can crystallize under favorable conditions. Our original design intent is to use a freezing-thawing (FT) technique to induce the PVA crystallization while an external deformation force is imposed on the hydrogel (Fig. 1c). This would induce chain anisotropy into the hydrogel network, which we anticipate would lead to a programmable two-way shape memory behavior using the VPT as the driving transition. Such behavior would go beyond the use of PVA as the driving transition for the one-way shape memory effect[25].

Specifically, the hydrogel is stretched by 100% and frozen-thawed. Afterward, the stretching force is removed and the strain is partially fixed due to the PVA crystallization. Supplementary Fig. 1 shows that the strain fixation rate increases with the freeze-thaw cycles and reaches a plateau value of 45% after seven cycles, implying that the crystallization reaches maximum. The stretching programmed hydrogel shows anisotropic dimensional shrinking (Fig. 1d) upon heating, in contrast to the isotropic shrinking of the same hydrogel without programming. The network anisotropy of the hydrogel is confirmed by 2D SAXS (Supplementary Fig. 2). For quantification, we define anisometry as the ratio between length and width after shrinking from an initially square-shaped hydrogel. Accordingly, the anisometry is found to increase linearly with the initial programming strain (Fig. 1e).

The mechanical properties and swelling ratios of the hydrogels with various MBA and PVA contents are shown in Supplementary Figs. 3 to 5. Higher MBA content results in larger modulus and smaller elongation at break. On the other hand, the modulus and the strength of the hydrogels with higher PVA content have been largely enhanced after repeated freeze-thawing. For hydrogels with higher PVA content, the swelling ratio undergoes a more remarkable decrease. These results imply that higher PVA content would lead to more additional crosslinking points due to the crystallization of PVA upon freeze-thawing. This is also critical to the FFE behavior. The FFE transformation of the sample containing 10 wt% PVA is remarkable, while the samples containing 1 wt% PVA do not exhibit visible FFE transformation. Accordingly, the MBA is fixed at 0.25 wt% of the NIPAM monomer and the PVA concentration is fixed at 10 wt% for further investigations.

For the optimal composition of MBA (0.25%) and PVA (10%), the equilibrium swelling behaviors of the hydrogels with different freeze-thawing conditions and applied strains are presented in Supplementary Fig. 6. Comparison between FT0 (no freeze-thaw and no external strain) and FT 7 ISO (freeze-thaw seven times without external strain) suggest that freeze-thawing significantly decreases the equilibrium swelling at temperatures below the VPT of 32 °C. Further comparison between FT 7 ISO and FT 7 ANISO (freeze-thaw seven times with an external strain of 100%) suggests that the equilibrium swelling ratio is slightly reduced by the applied strain. This is likely due to the impact of the applied strain on the PVA crystallization, which affects the physical crosslinking density (thus the equilibrium swelling). Nevertheless, this impact is small (around 15%). The equilibrium swelling above the VPT is identical for all the above three samples, regardless of the freeze-thaw and applied strain. In addition, the swelling of the programmed hydrogel is fully reversible upon cyclic heating and cooling (Fig. 1f) across the VPT, implying the stability of the PVA crystals in the process.

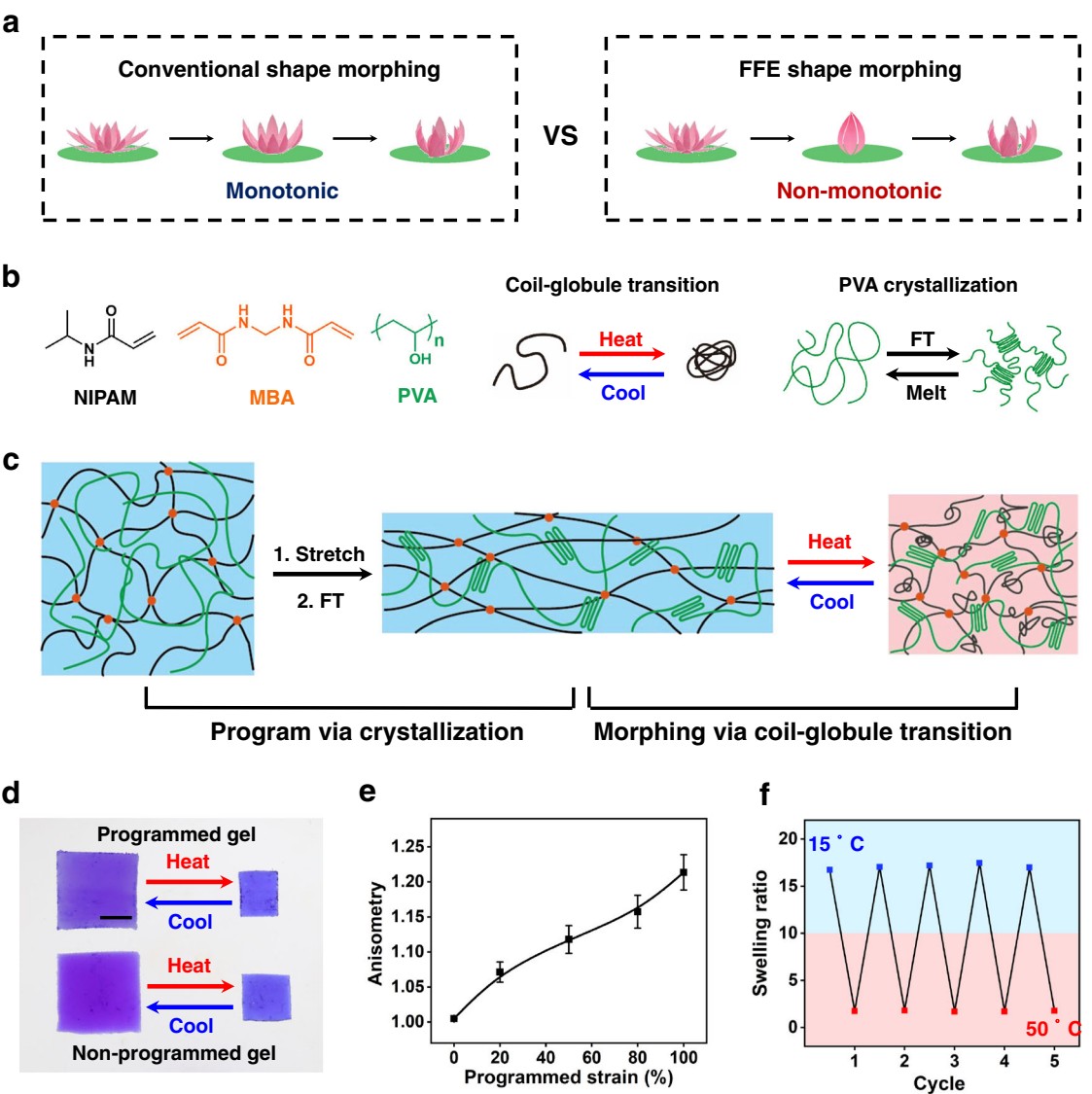

**Fig. 1 Chemical design and programming of the PNIPAM-PVA hydrogel. a** Schematic comparison between conventional and FFE shape-shifting behaviors. **b** Hydrogel composition, thermal transition, and PVA crystallization. **c** Mechanism for programming reversible shape-shifting. **d** Comparison of the shape-shifting behaviors for programmed and non-programmed hydrogels. Scale bar: 0.5 cm. **e** The impact of programmed strain on the anisometry of hydrogels. The error bars were obtained from three parallel experiments. **f** Reversibility of the programmed anisotropic hydrogels.

**FFE behavior of the hydrogels and the underlying mechanism**. When a rectangular gel is programmed into a zig-zag shape (Fig. 2a, equilibrium shape 1, ES1) by folding and freeze-thaw, an unexpected phenomenon occurs. Upon cooling, instead of direct expansion from ES1 to an equilibrium shape 2 (ES2), the gel (ES1) folds into tighter zig-zags and then unfolds to ES2 (Supplementary Movie 1). We call these intermediate tighter zig-zags the FFE shapes. With cutting and freeze-thaw under a different external force, the gel can be programmed to morph between two open four-arm grippers (ES1 and ES2 in Fig. 2b). Importantly, the gripper undergoes close–open dual action during the process due to the arising of the FFE shapes, despite the single stimulation (i.e., cooling). More complex dual actions operating are demonstrated in Supplementary Figs. 7 and 8, showing the versatility of the FFE programming method. The programmability and the non-monotonic nature of the shape-shifting stand in sharp contrast to that of conventional responsive hydrogels (Fig. 1a).

We resort to a simple bent shape as a model for revealing the mechanistic origin of the intriguing behavior. We speculate that

this phenomenon may be caused by two factors: geometry and stress. To understand this, two bent samples are fabricated by two different methods. The bent shape in Fig. 2c is made by cutting from a stress-free film whereas the one in Fig. 2d by mechanical programming from a rectangular film. Despite their nearly identical geometry at 50 °C, they are at very different stress states stress-free (Fig. 2c) versus stressed (Fig. 2d). The two samples show drastically different shape-shifting behavior upon cooling to 15 °C. The bent angle of 90° in Fig. 2c is reduced slightly to 80° during the transition to its expanded equilibrium shape (bent angle of 90°). In contrast, the bent angle of the stressed sample (Fig. 2d) is reduced from 90° to a full close state (bent angle of 0°), before it reopens to an equilibrium angle of 55°. Overall, although both samples show the close–open dual action, the degree of which is drastically different. This comparison suggests that geometry and stress both play a role, with the latter being the dominant contributor to the FFE behavior. We believe the geometric effect in Fig. 2c arises from the larger surface area on the outer side compared to the inner side around the bent area.

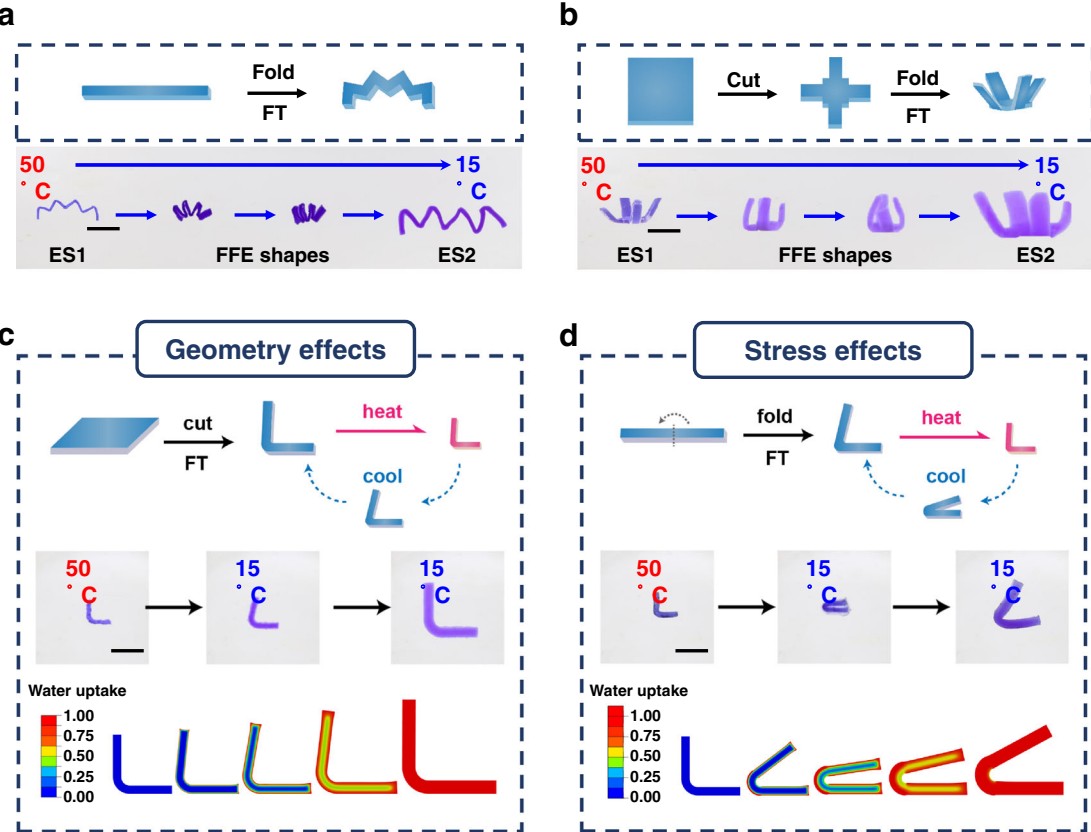

**Fig. 2 Demonstration of the FFE behaviors and the underlying mechanism. a**, **b** Programming and FFE behaviors of **a** a zig-zag and **b** a four-arm gripper. **c**, **d** Illustration, actual images, and FEA modeling of the shape-shifting process for **c** a stress-free hydrogel and **d** a stressed hydrogel. Scale bars are all 0.5 cm.

This causes faster swelling on the outer side, leading to a slight transient closing action. As the swelling process proceeds, water diffuses into the inner region of the gel, reducing the swelling ratio difference between the outer and inner sides, thereby giving rise to the reopen action. Indeed, this hypothesis is confirmed by the results of finite element analysis (FEA) modeling which demonstrates the close-reopen process of the stress-free sample induced by water diffusion (the bottom row of Fig. 2c).

The impact of stress is more complex. Typically, tension enhances hydrogel swelling while compression exhibits an opposite effect. The underlying principle is supported by a theoretical study from Hong et al.[26–28] Specifically, swelling of hydrogels is a diffusion process with the diffusion rate dictated by the water diffusivity $D$. In general, diffusivity is strongly affected by the stress state as follows: $D = D_0 e^{\sigma_h \Omega / k_B T}$, where $D_0$ is the diffusivity corresponding to zero stress, $\sigma_h$ the hydrostatic stress that can be obtained from the trace of the stress tensor, $\Omega$ is the activation volume of water diffusion, and $k_B T$ the thermal energy. From the equation, it can be concluded that tensile stress, which yields a positive $\sigma_h$, leads to a large diffusivity $D$ and more rapid swelling. On the contrary, compressive stress yields a negative $\sigma_h$, leads to a small diffusivity $D$ and slower water swelling.

For the current work, the programming process introduces tensile and compressive stresses on the outside and inside of the right angle, respectively. One possibility is that the differential stress causes water redistribution within the hydrogel itself that drives the FFE shape-shifting. We run a reference experiment in which the hydrogel is cooled in a silicone oil bath instead of water. No appreciable FFE phenomenon is observed, disproving this possibility. Another likely reason is that, at the early stage, the tensile stress induces faster water swelling on the outer side,

causing the drastic transient closing action. At a longer timescale, the inner side continues to swell due to its slower swelling rate while the outer side reaches the equilibrium state earlier. The result is that the shape reopens. The overall swelling kinetics of the anisotropic gel and the isotropic gel without programming is relatively close (Supplementary Fig. 9). However, our controlled experiments suggest that the anisotropic gel does indeed show a notably faster dimensional change in the stretching direction when compared to an isotropic gel without programming (Supplementary Fig. 10), confirming our proposed mechanism. The equilibrium folding angles of 55° (Fig. 2d) also deviates from the original angle of 90°. We believe this arises from the impact of stress on the equilibrium water uptake, consistent with literature reports[26–29]. FEA modeling (Fig. 2d) based on the above mechanisms agrees well with the experimental observations.

By monitoring the change in the folding angle, the kinetics of the FFE shape-shifting process is obtained (Fig. 3a). It is evident that the folding angle reduces quickly, reaching 0 at about 2.9 min. This fully folded state persists for 0.8 min, afterwards, it starts to unfold, reaching a plateau folding angle of 55° at 12.5 min. For comparison purposes, we fabricate a hydrogel (Supplementary Fig. 11) consisting of an identical PNIPAM-PVA layer and a poly(acrylamide) layer. This bilayer hydrogel can undergo shape-shifting in a conventional fashion (i.e., without the FFE). Figure 3b shows its shape-shifting kinetics as reflected in the time evolution of the geometric factors (defined in Supplementary Fig. 11). A close comparison between Fig. 3a and b clearly shows their drastic distinction (non-monotonic versus monotonic). Importantly, the timescale for the first FFE shape-shifting is 2.9 min (Fig. 3a), which is one order of magnitude shorter than the time corresponding to the turning point in

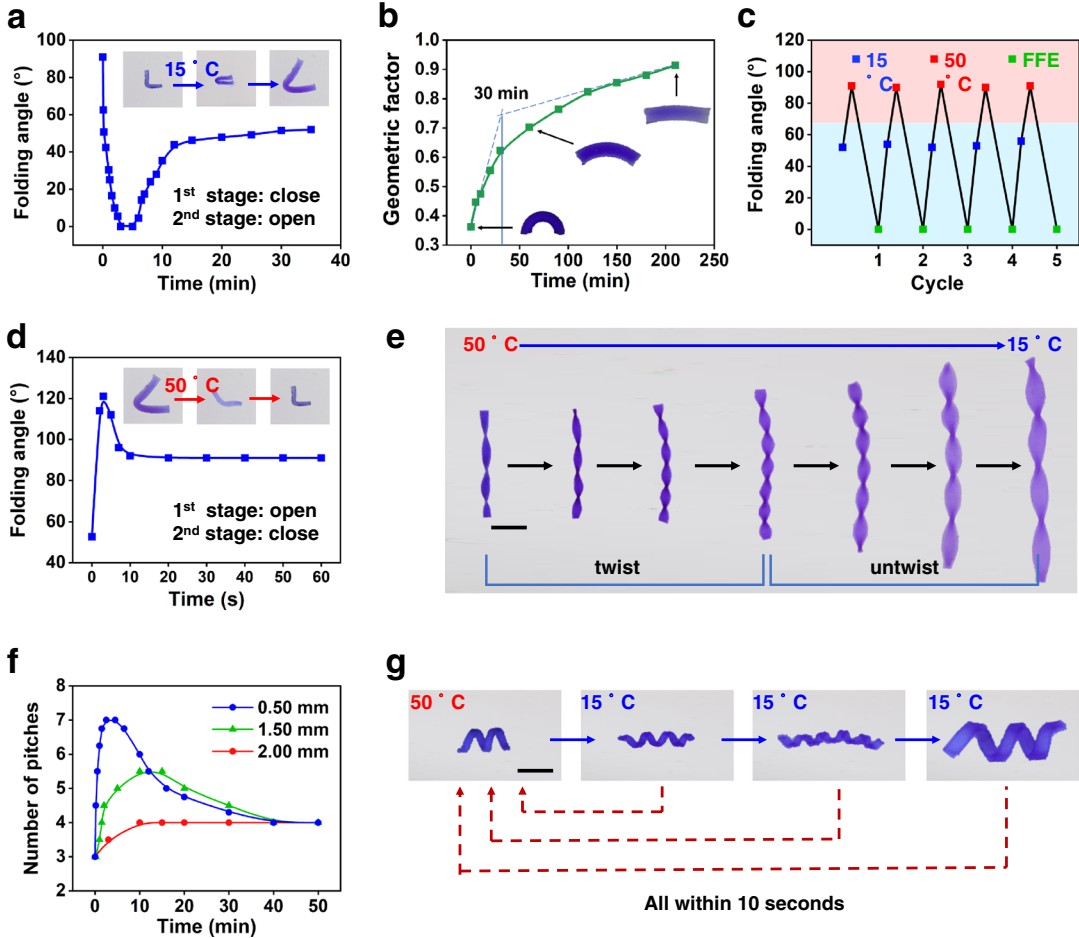

**Fig. 3 Kinetics and reversibility of the FFE shape-shifting process. a** FFE based shape-shifting kinetics upon cooling. **b** Conventional shape-shifting kinetics for a bilayer gel. **c** Reversibility of FFE shape-shifting. **d** FFE based shape-shifting kinetics upon heating. **e** Demonstration of the FFE based twist-untwist cycle. **f** Impact of the sample thickness on the twist-untwist shape-shifting. **g** Rapid reversion of the FFE shapes upon heating. Scale bar: 1 cm.

Fig. 3b. The results suggest that the FFE phenomenon not only diversifies the shape-shifting pathway but also markedly accelerates the process. This unique FFE shape-shifting behavior is highly reproducible, as evidenced in the cyclic tests in Fig. 3c. We note that the FFE behavior is not limited to the cooling process. In fact, during heating from 15 to 50 °C, we also observe that a non-monotonic open–close dual action occurring at a fast pace (within 10 s, Fig. 3d).

We next use a different geometry for further investigation, the twisted shape programmed from a rectangular flat strip (Fig. 3e). During the cooling process, the overall length increases continuously which is consistent with the continuous swelling. However, the geometric change undergoes an unusual twist-untwist dual action (Supplementary Movie 2). This FFE shape-shifting can be quantified by monitoring the time evolution of the number of pitches. Accordingly, Fig. 3f shows that the number of pitches increases sharply from 3 to 7 in the first 5.0 min and reduces at a slower pace afterward to 4 at 50 min. Because of the underlying diffusion-based mechanism, we anticipate that sample thickness has a strong impact on the phenomenon. Indeed, Fig. 3f shows that a thinner sample (0.50 mm) provides a notably more pronounced FFE behavior than a thicker sample (1.50 mm) and the behavior vanishes when the thickness reaches 2.00 mm. The width of the original rectangular shape also affects the extent of twisting/untwisting actions (Supplementary Fig. 12) because of its anticipated impact on the stress distribution. We note here that all of the FFE shapes can quickly morph back into the initial

equilibrium shape (within 10 s) when they are put back into a 50 °C water bath (Fig. 3g).

**Fabrication of FFE hydrogel devices**. Figure 4 illustrates conceptually how the FFE behavior can be harnessed for device applications beyond what is possible with conventional hydrogels. Specifically, we fabricate a device with eight straight arms which are subsequently programmed into partially folded forms (Fig. 4a). Upon cooling, the device shrinks first and expands afterward, showing a very large change in the device diameter (Fig. 4a, b). The device is placed above the top hole on a box (Supplementary Fig. 13). Herein, the hole diameter of 2.1 cm is notably smaller than the diameters of the two equilibrium shapes (4 and 5.8 cm, respectively). Despite that, the device upon cooling shrinks, drops into the box, and expands (Fig. 4c, Supplementary Fig. 14, and Supplementary Movie 3). This dual opposite action triggered by a single stimulus (cooling) is otherwise impossible with conventional hydrogels. The behavior can be potentially harnessed for practical applications such as minimally invasive medical devices. In a situation like that, access to two different stimuli is much harder than one stimulus. Being able to drive two opposite shape-shifting with one stimulus can open up new opportunities, in addition to larger and faster shape transformations mediated by the FFE shape-shifting mechanism.

The above FFE shape-shifting behavior driven by direct temperature change can be expanded to other hydrogel systems. When photothermal graphene oxide (GO) fillers are incorporated

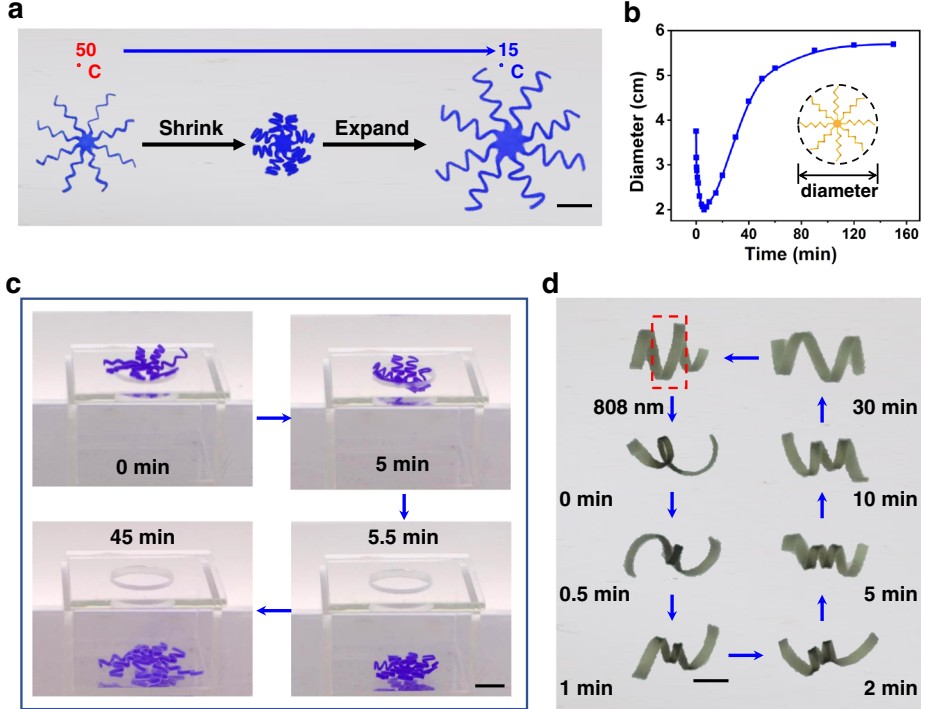

**Fig. 4 Demonstration of an eight-arm FFE shape-shifting device. a** Top view of the shrinking and subsequent expansion upon cooling. Scale bar: 1.0 cm. **b** Quantitative time evolution of the device diameter. **c** Dual opposite actions enabling the device to pass through the small hole. Scale bar: 1.0 cm. **d** Light triggered spatial actuation and the subsequent FFE transformation. The sample in cool water was exposed by near IR light (808 nm) at the framed location for 20 s. Subsequent photos at different times show the FFE behavior. Scale bar: 0.5 cm.

into the hydrogel, spatially controllable FFE shape-shifting can be realized with near IR light. The hydrogel actuator can respond rapidly to light exposure (Supplementary Fig. 15). As shown in Fig. 4d, upon localized exposure to near IR light for 20 s, the light was removed and the actuator underwent the complex FFE shape transformation during the subsequent reswelling. In addition to the thermo-responsive PNIPAM hydrogels, a pH-responsive hydrogel composed of polyacrylic acid (PAA) and PVA was prepared. After the mechanical programming process based on the freeze-thawing effect of PVA, the hydrogel exhibits a similar non-monotonic FFE shape-shifting upon pH change (Supplementary Figs. 16 and 17). The above two experiments prove that the unique FFE shape-shifting mechanism is general for responsive hydrogels.

Finally, we note that the use of the far-from-equilibrium (FFE) terminology to describe the observed unusual phenomena deserves careful considerations. Our initial thought was inspired by a previous review that described various existing FFE phenomena[30]. In that reference, earthquake and rapidly cooled metallic glass were used as two prime examples of FFE phenomena. In both these cases, the phenomena are equilibrium-driven over the long term. However, along the pathways, the systems evolve through thermomechanically unstable states that accumulate high energy, which would eventually be released if the timescale is sufficiently long. It is, however, these highly unstable thermodynamic states that give rise to the unusual phenomena, which is different from normal kinetic processes[30–32]. Our shape-shifting system behaves similarly. That is, instead of monotonic shifting between the two equilibrium states as is the case of typical hydrogels, its kinetics is such that it drives the pathway way beyond the normal pathway. In particular, the initial small difference in diffusion is enlarged by the resulting stress, which further leads to a greater difference in diffusion. This amplifying effect is particularly noteworthy.

Eventually, it leads to the unusual non-monotonic pathway at an accelerated pace. Although the molecular structure of the network evolves between different equilibrium states mediated by two-phase transitions, the differential swelling kinetics in different regions "unnaturally" creates internal osmotic pressure (energy) that leads to the unusual intermediate state (or shape) at the system level. We, therefore, use the term "FFE" to describe our system.

## Discussion

The FFE behavior illustrated above introduces a versatile dimension towards designing future shape-shifting materials. Although the current study focuses on hydrogels, the underlying diffusion-based mechanism is quite general. Its mild enabling conditions and kinetic acceleration are both attractive attributes for engineering applications. In particular, the mechanism stands in contrast to many natural FFE phenomena that require a lengthy accumulation of large energies under extraordinary conditions[28]. We believe that similar principles can be applicable to a wide variety of shape-shifting materials. Unusual benefits can be potentially realized for various devices such as vapor responsive soft robots, artificial muscles, minimal invasive medical devices, and high sensitivity sensors.

## Methods

**Materials**. *N*-Isopropylacrylamide (NIPAM) was purchased from TCI and purified by recrystallization with *n*-hexane. acrylic acid (AA), *N,N′*-Methylenebisacrylamide (MBA), and polyvinyl alcohol (PVA, $M_w = 89,000–98,000$, 99 + % hydrolyzed) were acquired from Sigma-Aldrich. Ammonium persulfate (APS), tetra-methylethylenediamine (TEMED) were procured from J&K. Graphene oxide (GO) dispersion (1–5 layers, 1 mg/mL water) was purchased from XFNANO Inc. All chemicals were used without further purification except NIPAM.

**Synthesis of hydrogels**. The FFE hydrogel was synthesized by free-radical copolymerization of NIPAM and MBA in PVA aqueous solution (10 wt%).

Specifically, 1.0 g of NIPAM and 0.25 wt% MBA were dissolved in 5 mL of 10 wt% PVA aqueous solution. After adding 200 μL of APS aqueous solution (4 wt%) and 40 μL of TEMED subsequently, the solution was quickly transferred into a cell composed of two glass slides with poly(dimethylsiloxane) rubber as the spacer. The reaction proceeded in a refrigerator at 4 °C for 24 h. The photothermally responsive hydrogel was synthesized according to the same approach as the FFE hydrogel, except that 0.1 wt% GO is added into the precursor solution. The pH-responsive hydrogel was synthesized as follows: 1.0 g of AA and 0.25 wt% MBA were dissolved in 5 mL of 10 wt% PVA aqueous solution. After adding 100 μL of APS aqueous solution (4 wt%), the solution was quickly transferred into a cell composed of two glass slides with poly(dimethylsiloxane) rubber as the spacer. The reaction proceeded at 70 °C for 12 h.

After the polymerization, all the gels were incubated in deionized water for 1 day to remove the unreacted monomers.

**Measurements of swelling ratio**. The swelling ratios were calculated as $W_s/W_d$, where $W_s$ was the weight of the swollen gel and $W_d$ was the weight of the sample in the dry state. The latter was determined by incubating the sample in a 70 °C oven for 1 day to ensure that the water was completely removed. The swelling ratios at different temperatures were measured after equilibrating the samples at the corresponding temperature for 6 h.

**Programming process**. The programming process was implemented according to the following steps. An external force was applied to deform the sample (e.g., bending, stretching, and twisting), followed by repeated freezing-thawing operations several times with the deformation force maintained throughout the process. The freezing temperature and melting temperatures were −20 and 15 °C, respectively. The freezing time was 6 h and the melting time was 30 min. Upon completion of the freezing-thawing operations, the sample was equilibrated in 15 °C water for 1 day before further evaluation of the shape-shifting behaviors.

**Measurements of fix ratios and anisometry**. To determine the shape fix ratio, a rectangular sample (20 mm × 2 mm × 1 mm) was stretched 100% and underwent freezing-thawing seven times. A fixity ratio was calculated as

$$R_f = \frac{l - l_0}{l_0} \times 100\%$$

where $l_0$ and $l$ represent the sample length before and after the fixation process, respectively.

To determine the gel anisometry, samples fixed to different strains by the freezing-thawing process were cut into square films (25 mm × 25 mm). The sample dimensions were measured after equilibration at 50 °C water. The anisometry was calculated as the ratio between the lengths parallel and perpendicular to the stretching direction.

**Finite element modeling**. The thermal diffusion and expansion model in FEM package ABAQUS was used to simulate the shape-shifting process of hydrogels, because the governing equations for mass diffusion and associated volume change take the same form as those for thermal diffusion and expansion (Supplementary Table 1). A neo-Hookean hyperelastic model with changing modulus was applied to model the swelling of PNIPAM-PVA hydrogels. The hyperelastic potential is $W = \frac{E}{6}(I_1 - 3)$, with $E$ being Young's modulus of the hydrogel and $I_1$ the first invariant of deformation gradient tensor. In the modeling, the swelling-induced volume expansion was modeled via a thermal-expansion analogy, such that the linear swelling strain was given by $\alpha_v \Delta T(X)$, where $\alpha_v$ is the thermal-expansion coefficient, $\Delta T(X)$ is the temperature change set in the simulation which represents the water concentration at a material element of coordinates $X$ in the hydrogel. The swelling ratio of the hydrogel, $\beta_v$, was thus given by $\beta_v = e^{3\alpha_v \Delta T(X)}$. In the modeling, we take $\Delta T = 1$ to represent the equilibrium water concentration of a stress-free PNIPAM-PVA hydrogel. $\beta_v$ was experimentally measured to be 8.26 for a stress-free hydrogel and hence the value of $\alpha_v$ used in the simulation was evaluated as $\alpha_v = \frac{1}{3} \ln \beta_v = 0.70$. The Young's modulus $E$ decreased as the gel swelled and was given by $E = E_0 e^{-\alpha_v \Delta T}$, where $E_0 = 129 \, kPa$ denoting Young's modulus of the as-made gel was measured experimentally. To model the swelling process of PNIPAM-PVA hydrogels, for samples without prestress, we set $\Delta T = 1$ on the entire outer gel surface; for prestressed samples, we set $\Delta T = 1.6$ and 0.8 on the outer and inner sides of the bent region, respectively, and $\Delta T = 1$ on the rest part of the surface. Notably, although $\Delta T$ represents temperature change in the equation used in the modeling, in reality, it corresponds to the variation of water concentration but not the temperature change. For the modeling, we set different $\Delta Ts$ and a constant $\alpha_v$ for the two surfaces to be consistent with the thermal diffusion and expansion. That is, $\alpha_v$ is set to be a material constant.

**Small-angle X-ray scattering (SAXS) analysis**. SAXS analysis was performed on a Nano-inXider vertical SAXS/WAXS system (Xenocs, Sassenage, France) equipped with a semiconductor pixel detector (Pilatus 200 K, Dectris, Swiss). In this configuration, the sample-to-detector distance was 936 mm. The medium-resolution values for the setup results in the effective $q$ ranges for SAXS is from 0.09 to 4.49 nm$^{-1}$. Here, $q$ denotes the magnitude of the scattering vector $q$, with $q = 4\pi \sin \theta/\lambda$, where $\theta$ is half the scattering angle and $\lambda$ is the X-ray wavelength (0.154 nm). The swollen hydrogel sample was loaded into a cell covered by the X-ray transparent polyimide film, then fastened on a rotatable sample stage. The stage was rotated around the X-ray beam during sequential exposures. The final 2D SAXS patterns with the azimuthal coverage of 270° were displayed by the automatic merge of the acquired images. All data were corrected by the background and air scattering.

## Data availability
The data that support the findings of this study are available from the corresponding author on reasonable request.

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

## Acknowledgements

We would like to thank the following programs for their financial support: National Natural Science Foundation of China (grant nos. 51822307, 52033009, and 21625402).

## Author contributions

T.X. and Q.Z. conceived the concept and directed the project. Y.Z. designed and conducted the experiments with assistance from K.L., C.N., and D.C. and C.L. Y.Z., J.Z., and P.P. conducted and analyzed the SAXS characterization. Z.J., T.L., Y.Z., and J.G. conducted the FEA modeling. Y.Z., Q.Z., and T.X. wrote the paper with input from Z.J. All authors analyzed and interpreted data

## Competing interests

The authors declare no competing interests.
