## [Peer Review File · Nature Communications]

Differential diffusion driven far-from-equilibrium
shape-shifting of hydrogelsREVIEWER COMMENTS

Reviewer #1 (Remarks to the Author):

The paper by Zhang et al describes multishape memory transformation of a hybrid polymer network composed of two different polymer strands: thermo-sensitive PNIPAM and crystallizable PVA. Due to the distinct mechanisms of PNIPAM and PVA phase transitions, these networks allow separately shape programming and shape shifting. In addition, PVA crystallization during the programming step provides anisotropic pathway for shape-shifting, while heterogeneous swelling of the PNIPAM sub-network creates bending. The combination of these features allows generating an array of interesting shape transformations presented in Figures 2, 3, and 4. From this angle, the paper may deserve publication. However, prior to publication, major revisions are required.

1. The main issue is that the paper is highly descriptive. Interesting shape transformations are presented without any supporting study of structure-property correlations that cause these transitions.

- Dynamic mechanical studies should be conducted for both as prepared and programmed shapes to monitor phase transformations of both PNIPAM and PVA components (onset time / temperature, and transition shape / magnitude). Correlations between mechanical properties and network composition and crosslink density should be analyzed as they are directly related to shape transformations.

- Swelling behavior as a function of gel composition and externally applied strain should be studied as well. An equilibrium swelling ratio depends on many internal and external factors. It is very well-known that gel extension enhances swelling, while gel compression drives the solvent out of gel. This occurs due the deformation caused change of the balance between osmotic pressure and entropic chain elasticity. Therefore, the observed bending is trivial. However, its magnitude and speed depend on network structure, applied strain, and sample dimensions. The bending speed is controlled by swelling rate, which depends on sample size as r^2 . Smaller/thinner samples swell much faster. However, bending of thinner samples results in more homogeneous strain distribution, which is a driving force for bending. An interplay of these effects should be studied.

2. The other big issue of this paper is that it introduces many misleading concepts and terminologies.

- The observed shape-shifting behavior is nothing to do with the far-from-equilibrium (FFE) thermodynamics. Even though the shape transformations look dis-continuous, molecular structure of the studied networks evolves between different equilibrium states mediated by two phase transitions.

- Self-driven and self-amplifying are also incorrect. The observed shape alterations are externally driven by heating/cooling and controlled by externally applied and quenched strain distribution. There is neither internal energy source nor molecular reactions involved in the observed shape transformations. This is also related to the FFE thermodynamics, which usually assumes a continuous flux of energy and matter.

- Morphing has much wider meaning than shape-shifting, which is misleading in many

- The definition of anisotropy in line 100 is odd. Anisotropy is usually used in relation to direction-dependent properties and structures. Uneven shapes are usually called anisometric.

Reviewer #2 (Remarks to the Author):

The authors present a temperature-responsive hydrogel that deforms non-monotonically in response to a temperature change. The gel is composed of a PNIPAM polymer network and a solvent of water and PVA chains. After synthesis, the gel is deformed, the PVA chains are crystallized by freeze-thaw, and the gel is released. The resulting anisotropy and residual stress cause stretched regions to swell more quickly than compressed regions. Since the equilibrium swelling ratio is similar for both regions, the equilibrium shape is similar to the unswollen shape.

The combination of PNIPAM and PVA is unusual and produces intriguing phenomena, and the experiments performed here elucidate the mechanism for it. The results should definitely be published somewhere. This reviewer hesitates to recommend the publication in Nature Communications for two reasons. First, it is unclear how general the presented ideas are in

materials science. Second, it is unclear how useful the presented ideas are in practice. This reviewer will be delighted to reconsider this manuscript if the authors can make convincing arguments about these two general concerns. A list of more specific comments follows.

1. The paper presents a kinetic phenomenon. The phrase "far-from-equilibrium" seems to be a distraction. All kinetic phenomena happen away from equilibrium. How far is far enough to be called far?
2. The argument for why the tensile stress on the outside corner of the bent gel causes more rapid swelling in this region seems unclear. Please explain further.
3. Lines 150-153, the percentage length change in the linear case should be stated explicitly in the main text. For example, I calculate an approximately ~50% faster length change in the anisotropic gel than in the isotropic gel. Comparing arc lengths of the inner and outer radii of the corner of the bent gel, does this length change correspond to the difference in bending angle that is observed?
4. Was the swelling ratio as a function of time of the anisotropic gel different from the isotropic gel measured? Similar to Supplementary Figure 5, but for swelling ratio instead of length. If so, then this would cause β_V in the FEA to be different for the two surfaces.
5. The FEA analysis models the difference in swelling ratio for the anisotropic gel as a difference in the change in temperature (lines 270-272, "for pre-stressed samples, we set $\Delta T = 1.6$ and 0.8 on the outer and inner sides of the bent region, respectively, and $\Delta T = 1$ on the rest part of the surface."). Since the temperature change is the same for the entire gel in the experiment, why was α_V not changed for the different surfaces? Supplementary figure 3 shows that the swelling ratios are not equal. Also see comment 3.
6. This is a kinetic paper. How does the FEA analysis consider the kinetics of the shape change? The methods section does not mention water or heat diffusion, although some kinetic law must be implemented to generate the time-dependent results shown in Figure 2. The package used should be specified as well, since it may account for these. Currently, the methods section only states "A commercial FEM package ABAQUS was used to simulate the shape-morphing process of our system."
7. In the conclusion, the authors mention that the mechanism is quite general, but they do not mention examples of other materials systems that could be used.

Reviewers' comments:

Reviewer #1 (Remarks to the Author):

The paper by Zhang et al describes multishape memory transformation of a hybrid polymer network composed of two different polymer strands: thermo-sensitive PNIPAM and crystallizable PVA. Due to the distinct mechanisms of PNIPAM and PVA phase transitions, these networks allow separately shape programming and shape shifting. In addition, PVA crystallization during the programming step provides anisotropic pathway for shape-shifting, while heterogeneous swelling of the PNIPAM sub-network creates bending. The combination of these features allows generating an array of interesting shape transformations presented in Figures 2, 3, and 4. From this angle, the paper may deserve publication. However, prior to publication, major revisions are required.

Response: We sincerely thank the reviewer for the professional feedback on our work.

1. The main issue is that the paper is highly descriptive. Interesting shape transformations are presented without any supporting study of structure-property correlations that cause these transitions.

Response: We thank the reviewer for this comment which helps to improve our manuscript. The structure-property correlations have been further studied and discussed based on additional experiments as follows.

- Dynamic mechanical studies should be conducted for both as prepared and programmed shapes to monitor phase transformations of both PNIPAM and PVA components (onset time / temperature, and transition shape / magnitude). Correlations between mechanical properties and network composition and crosslink density should be analyzed as they are directly related to shape transformations.

Response: Mechanical properties of the hydrogels with different compositions and crosslinking densities during the repeated freeze-thawing cycles have been characterized in the revision. The results are presented in Supplementary Figures 1 and 2 (as well as shown below). Higher MBA content results in larger modulus and smaller elongation at break (Supplementary Figures 1). Hydrogels with lower MBA contents exhibit more remarkable improvement in the mechanical properties. Therefore, MBA content of 0.25 wt% of the NIPAM monomer was chosen for further investigations. Before the freeze-thawing, the PVA composition has a slight influence on the mechanical properties. After the freeze-thawing, the modulus and the strength of the hydrogels with higher PVA content have been greatly enhanced (Supplementary Figures 2). On the other hand, the swelling ratio of the hydrogels with repeated freeze-thawing times have also been monitored as shown in Supplementary Figure 3a (as well as shown below). For hydrogels with higher PVA content, the swelling ratio undergoes more remarkable decrease. In comparison, mechanical properties and swelling ratio of hydrogels with a low PVA content exhibit

no obvious change upon repeated freeze-thawing. The results implies that higher PVA content would lead to more additional physical crosslinking points due to the crystallization of PVA upon freeze-thawing. Accordingly, the PVA concentration was fixed at 10 wt% for further investigations.

The samples with different PVA contents exhibit different initial angles when subjected to identical crystallization-induced programming. More specifically, increasing the PVA content can improve the shape fixity ratio during initial programming (Supplementary Figures 3b). Further from this figure, it can be seen that the sample containing 10 wt% PVA exhibits an obvious FFE transformation, while the FFE transformation of the sample containing 5 wt% PVA is notably weakened and the samples with 1 wt% PVA does not exhibit FFE transformation.

Corresponding changes have been made in Page 4 in the revision.

Supplementary Figure 1. Tensile test of the hydrogels prepared from various MBA content (0.10 wt%, 0.25 wt%, and 0.50 wt% of the NIPAM monomer) at a constant concentration of the PVA solution (10 wt%) before (a) and after freeze-thawing (b).

Supplementary Figure 2. Tensile test of the hydrogels prepared from PVA solution with various concentration (1 wt%, 5 wt%, and 10 wt%) at a constant MBA feed (0.25 wt% of the NIPAM monomer) before (a) and after freeze-thawing (b).

Supplementary Figures 3. Effect of PVA content on swelling ratio and the FFE behavior. (a) Swelling ratios of hydrogels with different PVA content under freeze-thawing cycles; (b) Kinetics of the FFE morphing process upon cooling for hydrogels with different PVA after freeze-thawing for 7 cycles.

- Swelling behavior as a function of gel composition and externally applied strain should be studied as well. An equilibrium swelling ratio depends on many internal and external factors. It is very well-known that gel extension enhances swelling, while gel compression drives the solvent out of gel. This occurs due to the deformation caused by change of the balance between osmotic pressure and entropic chain elasticity. Therefore, the observed bending is trivial. However, its magnitude and speed depend on network structure, applied strain, and sample dimensions. The bending speed is controlled by swelling rate, which depends on sample size as r^2 . Smaller/thinner samples swell much faster. However, bending of thinner samples results in more homogeneous strain distribution, which is a driving force for bending. An interplay of these effects should be studied.

Response: See the response to the previous comment above, the impact of network structure (MBA and PVA contents) on the swelling and FFE shape-shifting behavior is studied and discussed.

For the optimal composition of MBA (0.25%) and PVA (10%), the equilibrium swelling behaviors of the hydrogels with different freeze-thawing conditions and applied strains are presented in Supplementary Figure 6. Comparison between FT0 (no freeze-thaw and no external strain) and FT 7 ISO (freeze-thaw seven times without external strain) suggests that freeze-thawing significantly decreases the equilibrium swelling at temperatures below the VPT of 32 °C. Further comparison between FT 7 ISO and FT 7 ANISO (freeze-thaw seven times with an external strain of 100%) suggests that the equilibrium swelling ratio is slightly reduced by the applied strain. This is likely due to the impact of the applied strain on the PVA crystallization, which affects the physical crosslinking density (thus the equilibrium swelling). Nevertheless, this impact is small (around 15%). In addition, the equilibrium swelling above the VPT is identical for all the above three samples, regardless of the freeze-thaw and applied strain. Corresponding changes have been made on Page 5 and 6 in the revision.

As for the impact of thickness on shape-shifting speed and magnitude, the results are presented in Figure 3f and discussed on the corresponding main text. A thinner sample (0.50 mm) provides a notably more pronounced FFE behavior than a thicker sample (1.50 mm) and the behavior vanishes when the thickness reaches 2.00 mm.

Supplementary Figure 6. Swelling ratios of the hydrogels at various temperature (Black and red lines represent samples with 0 and 7 times freezing-thawing, respectively; blue lines represent an anisotropic sample with 100% programming strain and 7 times freezing-thawing). All three curves show transitions around 32 °C.

2. The other big issue of this paper is that it introduces many misleading concepts and terminologies.

- The observed shape-shifting behavior is nothing to do with the far-from-equilibrium (FFE) thermodynamics. Even though the shape transformations look dis-continuous, molecular structure of the studied networks evolves between different equilibrium states mediated by two phase transitions.

Response: We agree that use of these terminologies deserves more careful considerations. Our initial justification of the term of far-from-equilibrium (FFE) came from reference 28. In this reference, earthquake and rapidly cooled metallic glass are used as two prime examples of FFE phenomena. In both these cases, the phenomena are equilibrium driven over the long term as is the case for everything on Earth. However, along the pathways, the systems evolve through a thermomechanically unstable states that accumulates high energy, which would eventually be released. It is these highly unstable thermodynamic states that give rise to the unusual phenomena, which is different from normal kinetic processes. Our shape-shifting system behaves similarly. That is, instead of monotonic shifting between the two equilibrium states as is the case of typical hydrogels, its kinetics is such that it drives the pathway way beyond the normal pathway. In particular, the initial small difference in diffusion is enlarged by the resulting stress, which further

leads to greater difference in diffusion. This amplifying effect is particularly noteworthy. Eventually, it leads to the unusual non-monotonic pathway at an accelerated pace. We fully agree that the molecular structure of the networks evolves between different equilibrium states mediated by two phase transitions. However, at the system level, the differential swelling kinetics in different regions “unnaturally” creates internal osmotic pressure (energy) that leads to the unusual intermediate state (or shape). We therefore consider the term “FFE” appropriate for our system. Nevertheless, the above discussion is provided on Page 11 and 12 of the revised manuscript for clarification.

- Self-driven and self-amplifying are also incorrect. The observed shape alterations are externally driven by heating/cooling and controlled by externally applied and quenched strain distribution. There is neither internal energy source nor molecular reactions involved in the observed shape transformations. This is also related to the FFE thermodynamics, which usually assumes a continuous flux of energy and matter.

Response: Agreed. “self-driven” and “self-amplified” are removed from the title and main text.

- Morphing has much wider meaning than shape-shifting, which is misleading in many

Response: The word “morphing” has been changed into “shape-shifting”.

- The definition of anisotropy in line 100 is odd. Anisotropy is usually used in relation to direction-dependent properties and structures. Uneven shapes are usually called anisometric.

Response: We thank the reviewer for this suggestion. The word “anisotropy” has been changed into “anisometry” in corresponding places.

Reviewer #2 (Remarks to the Author):

The authors present a temperature-responsive hydrogel that deforms non-monotonically in response to a temperature change. The gel is composed of a PNIPAM polymer network and a solvent of water and PVA chains. After synthesis, the gel is deformed, the PVA chains are crystallized by freeze-thaw, and the gel is released. The resulting anisotropy and residual stress cause stretched regions to swell more quickly than compressed regions. Since the equilibrium swelling ratio is similar for both regions, the equilibrium shape is similar to the unswollen shape.

The combination of PNIPAM and PVA is unusual and produces intriguing phenomena, and the experiments performed here elucidate the mechanism for it. The results should definitely be published somewhere. This reviewer hesitates to recommend the publication in Nature Communications for two reasons. First, it is unclear how general the presented ideas are in materials science. Second, it is unclear

how useful the presented ideas are in practice. This reviewer will be delighted to reconsider this manuscript if the authors can make convincing arguments about these two general concerns. A list of more specific comments follows.

Response: We sincerely thank the reviewer for the constructive comments. Additional experiments have been conducted and more indepth discussion provided.

In the original manuscript, we have presented that the thermo-responsive PNIPAM hydrogels with the non-monotonic shape-shifting behavior after the mechanical programming. In view of the reviewer's comment on the generality, we further designed and synthesized a pH responsive hydrogel composed of polyacrylic acid (PAA) and PVA. After the mechanical programming, the hydrogel was found to exhibit similar non-monotonic shape-shifting upon pH change (as shown in Supplementary Figures 16 and 17 below). In addition, we incorporated photothermal graphene oxide fillers into the thermos-responsive hydrogel to fabricate a light-triggered actuator. As shown in Fig. 4d, upon exposure to near IR light for 20 s, the light was removed and actuator underwent the FFE transformation during the subsequent reswelling. The above additional experiments prove that the FFE phenomenon is general. We've observed similar phenomena for solvent responsive polymers and the results will be presented in a separate study in the future.

As for the usefulness, the dual shape-shifting actions triggered by a single stimulus (cooling) is an important expansion beyond with conventional responsive hydrogels. Specifically, the demonstration in Figure 4 provides a hint on what is possible. We believe that the behavior can be potentially harnessed for practical applications such as minimal invasive medical devices. In a situation like that, access to two different stimuli is much harder than one stimulus. In addition, the FFE shape-shifting leads to larger and faster shape transformation, which could also be useful for practical situations that require that.

These two points have been clarified in the revision (Page 10 and 11).

Supplementary Figure 16. pH Responsive hydrogels. (a) Chemical structure of the precursors; (b) Deswelling kinetics of the hydrogels upon pH change from 13 to 1; (c) Swelling kinetics upon pH change from 1 to 13.

Supplementary Figure 17. The FFE transformation of the pH responsive hydrogels. (a) Schematic illustration of sample preparation. (b and c) Quantitative characterization and photographs of the twisting samples from pH=1 to pH=13. (d and e) Quantitative characterization and photographs of the twisting samples from pH=1 to pH=13 and then back to pH=1.

Figure 4d. Light triggered spatial actuation and the subsequent FFE transformation. The sample in cool water was exposed by near IR light (808 nm) at the framed location for 20 s. Subsequent photos at different times show the FFE behavior. Scale bar: 0.5 cm.

1. The paper presents a kinetic phenomenon. The phrase “far-from-equilibrium” seems to be a distraction. All kinetic phenomena happen away from equilibrium. How

far is far enough to be called far?

Response: All shape-shifting behaviors are kinetics-based including the one described here. However, the behavior described in the current work is fundamentally different from commonly known shape-shifting hydrogels in terms of the unexpected pathways and the underlying mechanism. In particular, the initial small difference in diffusion is enlarged by the stress, which further results in greater difference in diffusion. This amplifying effect is particularly noteworthy. We therefore use “far-from-equilibrium” for clear distinction. Detailed justification behind the term and corresponding revision can be found in the response to a comment by Reviewer #1. The question of “how far” is “far” is judged by the fundamentally different mechanism and the previously unknown shape-shifting behavior.

2. The argument for why the tensile stress on the outside corner of the bent gel causes more rapid swelling in this region seems unclear. Please explain further.

Response: The following discussion is provided on Page 8 for further clarification. “The underlying principle is supported by the theoretical study from Hong et al.²⁶ Specifically, swelling of hydrogels is a diffusion process with the diffusion rate dictated by the water diffusivity D . In general, diffusivity is strongly affected by the stress state as follows: $D = D_0 e^{\sigma_h \Omega / k_B T}$, where D_0 is the diffusivity corresponding to zero stress, σ_h the hydrostatic stress that can be obtained from the trace of the stress tensor, Ω is the activation volume of water diffusion, and $k_B T$ the thermal energy. From the equation, it can be concluded that a tensile stress, which yields a positive σ_h , leads to a large diffusivity D and more rapid swelling. On the contrary, a compressive stress yields a negative σ_h , leads to a small diffusivity D and slower water swelling.”

3. Lines 150-153, the percentage length change in the linear case should be stated explicitly in the main text. For example, I calculate an approximately ~50% faster length change in the anisotropic gel than in the isotropic gel. Comparing arc lengths of the inner and outer radii of the corner of the bent gel, does this length change correspond to the difference in bending angle that is observed?

Response: The percentage length change varies with time, thus it is impossible to use a single number to describe it. However, the corresponding curve (supplementary Figure 10) is provided for calculating the percentage length change at any time during the swelling.

For the bent gel, the stress distribution from the inside to the outside is continuous and uneven, which is fundamentally different from linear stretching. The outermost length change can be measured, but the bending angle is determined by various factors including internal stress distribution, water swelling degree, and gel thickness. This is more complex than what one would otherwise expect. We plan to establish a mechanical model and publishing a separate study on this in the future.

4. Was the swelling ratio as a function of time of the anisotropic gel different from the isotropic gel measured? Similar to Supplementary Figure 5, but for swelling ratio

instead of length. If so, then this would cause β_V in the FEA to be different for the two surfaces.

Response: In the FEA, β_V is set to be different for the two surface. Theoretically, tensile strain promote the swelling and compressive strain restrict the swelling as explained above (response to the comment 2 from the Review #2). Therefore, β_V for the outside is set to be larger than that for the inside surface.

The swelling ratio as a function of time of the isotropic and the anisotropic gels has been measured (Supplementary Figure 9). The volumic swelling kinetics of the two gels exhibits no remarkable difference at the initial 30 min when the FFE transformation happens, which is in sharp contrast with the kinetics of length change. Upon further swelling, the swelling ratio of the anisotropic gel is smaller than the isotropic gel. Such a results is consistent with the equilibrium swelling ratio as shown in Supplementary Figure 6, but seems to be contradict with the theory that tensile strain promote the swelling. This may be attributed to complicity of our hydrogel system. The PVA component will crystalize upon freeze-thawing. Anisotropy in the molecular structure upon deformation would lead to a larger crystallinity which reduces the swelling ratio in comparison to the isotropic sample. Nevertheless, the two surfaces are both undergone deformation and thus are both anisotropic sample. So, we consider that the different setting of β_V should be reasonable.

Supplementary Figure 9. Swelling kinetics of as-prepared, anisotropic (programmed strain: 100%) and isotropic gels.

5. The FEA analysis models the difference in swelling ratio for the anisotropic gel as a difference in the change in temperature (lines 270-272, “for pre-stressed samples, we set $\Delta T=1.6$ and 0.8 on the outer and inner sides of the bent region, respectively, and $\Delta T=1$ on the rest part of the surface.”). Since the temperature change is the same for the entire gel in the experiment, why was α_V not changed for the different surfaces? Supplementary figure 3 shows that the swelling ratios are not equal. Also see comment 3.

Response: In this work we use the thermal-expansion model instead of the mass diffusion model to simulate the water-diffusion process in the hydrogel, and use the

thermal-expansion strain to model the water-diffusion-induced strain. This is because that the former one has already been implemented in the commercial FEM package ABAQUS, whereas the mass diffusion model is not. The governing equation and formula of volumetric strain of mass diffusion take nearly the same form as those of thermal expansion as shown in Table R1. Such a treatment has been commonly applied for simulation of mass diffusion. Notably, ΔT corresponds to the variation of water concentration but not the temperature change in the experiment, while it represents the temperature change in the equation used in the modelling. For the modelling, we set different ΔT s and a constant α_v for the two surfaces since it is coincident with the thermal-expansion model. That is, α_v is a material constant and volume change of the material is caused by temperature change. The review is also quite right that from the experiment standpoint, ΔT should be a constant whereas α_v should be varied for the two surfaces. We make no disagreement with the reviewer. The purpose is both at resulting different β_v (representing the volume) for the two surfaces. Corresponding changes have been made on Page 15 for clarification.

	Mass Diffusion	Thermal Diffusion
Governing equation	Kinetics: $\vec{j} = -D\vec{\nabla}C$ Equilibrium: $\frac{\partial C}{\partial t} = \vec{\nabla}(D\vec{\nabla}C)$ C is the concentration. D is the diffusion coefficient.	Kinetics: $\vec{q} = -k\vec{\nabla}T$ Equilibrium: $\frac{\partial T}{\partial t} = \vec{\nabla}\left(\frac{k}{\rho c}\vec{\nabla}T\right)$ k is the thermal conductivity, ρ is the density and c is the specific heat.
Volumetric strain	$\gamma_v \Delta C$ γ_v is the diffusion-induced expansion coefficient	$\alpha_v \Delta T$ α_v is the thermal expansion coefficient

Table R1. Comparison between mass-diffusion model and thermal-diffusion model

6. This is a kinetic paper. How does the FEA analysis consider the kinetics of the shape change? The methods section does not mention water or heat diffusion, although some kinetic law must be implemented to generate the time-dependent results shown in Figure 2. The package used should be specified as well, since it may account for these. Currently, the methods section only states “A commercial FEM package ABAQUS was used to simulate the shape-morphing process of our system.”

Response: As discussed above, we use the thermal-expansion model instead of the mass diffusion model to simulate the water-diffusion process due to the implementation of the software ABAQUS. The water diffusion kinetics and the associated time-dependent shape change is considered by using the thermal diffusion and expansion module in FEA the software.

In the revision, the original statement “A commercial FEM package ABAQUS was used to simulate the shape-morphing process of our system” have been changed to “The thermal diffusion and expansion model in commercial FEM package ABAQUS was used to simulate the shape-morphing process of hydrogels, because the governing equations for mass diffusion and associated volume change take the same form as those for thermal diffusion and expansion.”

7. In the conclusion, the authors mention that the mechanism is quite general, but they do not mention examples of other materials systems that could be used.

Response: Please refer to the response to the general comments above. In short, we had presented that the thermo-responsive PNIPAM hydrogels with the non-monotonic shape-shifting behavior after the mechanical programming in the original manuscript. We further designed and synthesized a pH responsive hydrogel composed of polyacrylic acid (PAA) and PVA in the revision. After the mechanical programming, the hydrogel was found to exhibit similar non-monotonic shape-shifting upon pH change. In addition, we incorporated photothermal graphene oxide fillers into the thermos-responsive hydrogel to fabricate a light-triggered actuator, which exhibited the FFE transformation. The above additional experiments prove that the FFE phenomenon is general.

REVIEWERS' COMMENTS

Reviewer #1 (Remarks to the Author):

The authors did a good job with revising the paper. It is now recommended for publication.

Reviewer #2 (Remarks to the Author):

The authors have addressed most of my concerns from the first round of review. In particular, the additional experiments with other polymers and actuation methods (pH and light) clearly demonstrate the generality of this system. I recommend the article be accepted after minor revisions. The following is a list of responses to the comments from the first round that I still have concerns about; the numbers here match the comment numbers from the first round.

1. The justification for calling this system "far-from-equilibrium" is still highly descriptive. In particular, the quantitative interpretation of the data requires no far-from-equilibrium physics, such as that presented in reference 28. Instead, the system is well described by assuming local equilibrium everywhere (as is done in most kinetic problems). If no far-from-equilibrium physics is used to analyze the system, then why is such a great emphasis placed on it being FFE? Can the label be justified simply because the phenomenon occurs away from the equilibrium state? It is difficult for the reviewer to assess this claim with only one general reference to far-from-equilibrium physics provided (reference 28).

2. The study in reference 26 models hydrogel swelling by assuming a constant diffusivity, D . The equation $D = D_0 \exp(\sigma_h \Omega / k_B T)$ in the revised text is not presented in the reference. The authors should provide a reference for this equation and an estimate of the magnitude of the change in diffusivity in the system studied here. Since diffusion time scales as $t \sim L^2/D$, can this change in diffusivity cause the difference in swelling time seen experimentally? Moreover, following reference 26, increased tensile stress decreases the chemical potential of water in a gel, increasing the flux of water into the gel. Therefore, even with a constant diffusivity, the chemical potential change can explain faster swelling in areas with higher tensile stress. Is this the correct mechanism?

5. The text on page 15 refers to Supplementary Table 1, but this table is missing in the supplementary information.

Reviewers' comments:

Reviewer #2 (Remarks to the Author):

The authors have addressed most of my concerns from the first round of review. In particular, the additional experiments with other polymers and actuation methods (pH and light) clearly demonstrate the generality of this system. I recommend the article be accepted after minor revisions. The following is a list of responses to the comments from the first round that I still have concerns about; the numbers here match the comment numbers from the first round.

Response: We sincerely thank the reviewer's reconsideration and the positive feedback on our revision.

1. The justification for calling this system "far-from-equilibrium" is still highly descriptive. In particular, the quantitative interpretation of the data requires no far-from-equilibrium physics, such as that presented in reference 28. Instead, the system is well described by assuming local equilibrium everywhere (as is done in most kinetic problems). If no far-from-equilibrium physics is used to analyze the system, then why is such a great emphasis placed on it being FFE? Can the label be justified simply because the phenomenon occurs away from the equilibrium state? It is difficult for the reviewer to assess this claim with only one general reference to far-from-equilibrium physics provided (reference 28).

Response:

First of all, there is no established frameworks for far from equilibrium physics, as stated in original Ref. 28 (line 2 of page 5): "One conceptual difficulty posed by systems far from equilibrium thus arises from the **absence of established theoretical frameworks**."

Secondly, Ref. 28 (the first three lines of the last paragraph on page 2) states "Far-from-equilibrium behavior ... corresponds to qualitatively different types of behavior and response, typically associated with crossing some threshold into a new regime." Our reported phenomenon meets this description perfectly.

Two additional references (Ref. 30 and 31) related to far-from-equilibrium materials are now cited in the revised manuscript, where interested readers can find more information consistent with the argument above. In addition, the last version of the manuscript has provided a thorough explanation on why we use the term far-from-equilibrium. This paragraph is copied below for quick referencing.

30. Srinivasarao, M., Iannacchione, G. S. & Parikh, A. N. Biologically inspired far-from-equilibrium materials. *MRS Bull.* 44, 91-95 (2019).

31. Bigioni, T. P., Lin, X. M., Nguyen, T. T., Corwin, E. I., Witten, T. A. & Jaeger, H. M. Kinetically driven self assembly of highly ordered nanoparticle monolayers. *Nat. Mater.* 5, 265-270 (2006)

Finally, we note that the use of the far-from-equilibrium (FFE) terminology to describe the observed unusual phenomena deserves careful considerations. Our initial thought was inspired by a previous review that described various existing FFE phenomena²⁸. In that reference, earthquake and rapidly cooled metallic glass were used as two prime examples of FFE phenomena. In both these cases, the phenomena are equilibrium driven over the long term. However, along the pathways, the systems evolve through a thermomechanically unstable states that accumulate high energy, which would eventually be released if the timescale is sufficiently long. It is, however, these highly unstable thermodynamic states that give rise to the unusual phenomena, which is different from normal kinetic processes. Our shape-shifting system behaves similarly. That is, instead of monotonic shifting between the two equilibrium states as is the case of typical hydrogels, its kinetics is such that it drives the pathway way beyond the normal pathway. In particular, the initial small difference in diffusion is enlarged by the resulting stress, which further leads to greater difference in diffusion. This amplifying effect is particularly noteworthy. Eventually, it leads to the unusual non-monotonic pathway at an accelerated pace. Although the molecular structure of the network evolves between different equilibrium states mediated by two phase transitions, the differential swelling kinetics in different regions “unnaturally” creates internal osmotic pressure (energy) that leads to the unusual intermediate state (or shape) at the system level. We therefore use the term “FFE” to describe our system.

2. The study in reference 26 models hydrogel swelling by assuming a constant diffusivity, D . The equation $D = D_0 \cdot \exp(\sigma_h \cdot \Omega / k_B T)$ in the revised text is not presented in the reference. The authors should provide a reference for this equation and an estimate of the magnitude of the change in diffusivity in the system studied here. Since diffusion time scales as $t \sim L^2/D$, can this change in diffusivity cause the difference in swelling time seen experimentally? Moreover, following reference 26, increased tensile stress decreases the chemical potential of water in a gel, increasing the flux of water into the gel. Therefore, even with a constant diffusivity, the chemical potential change can explain faster swelling in areas with higher tensile stress. Is this the correct mechanism?

Response:

Ref. 26 has established a generic theoretical framework for investigating the equilibrium swelling behavior of hydrogels subject to mechanical loads. By integrating the framework with kinetic diffusion laws, in another paper from the same authors (added as Ref. 27 in the revision), the thickness of a gel blanket under compressive stresses is plotted as a function of time with various levels of applied stresses. As evident in the plot, the higher the compressive stress (i.e., the more negative the value of $s\nu/kT$), the more the gel deswells and the faster the deswelling. For example, doubling the compressive stress $s\nu/kT$ from -0.05 to -0.1 reduces the normalized time ($\sqrt{tD/L^2}$) needed for reaching equilibrium from ~ 20 to ~ 12.5 , that is, the deswelling process is accelerated by a factor about 2.6. Likewise, gels subjected to higher tensile stresses will swells more and faster.

The thickness of the gel as a function of time

The effect of applied stress on accelerating the deswelling/swelling process of hydrogels mentioned above are a consequence of coupled network deformation, water concentration distribution, and chemical potential gradient (Ref. 26 and 27), which appears to be too complicated to understand for the majority of potential readers in soft materials and polymer chemistry. In this regard, we employ a phenomenological expression of effective diffusivity $D = D_0 e^{\sigma_h \Omega / k_B T}$, which has been widely used to describe diffusion under stresses, to qualitatively explain how stresses affect the swelling process (see additional Ref. 28). Here, D is an overall and apparent diffusivity. All the factors that affect the swelling kinetics (e.g. the stress and the chemical potential) have been taken into consideration. It should be noted that this expression is qualitatively right but quantitatively less precise than the results of the deduction presented in Ref. 27 due to the large deformation of hydrogels under external force. In experimental, however, it is yet difficult to precisely observe the difference in swelling time caused by various stress due to the complicity of our hydrogel system. Larger stress would lead to larger crystallinity as additional crosslinking points which reduces the overall swelling ratio.

Detailed studies in mechanics and modeling are of great interest. Indeed, we are finding a more simplified material system for further investigation, which belongs to relatively independent work. Therefore, the aforementioned discussion is not provided in the revision. Nevertheless, Ref. 27 and 28 have been added to the revision. Readers who are interested in the quantified study of the influence of external force on the swelling kinetics could refer to these literatures.

27. Hong, W., Zhao, X., Zhou, J. & Suo, Z. A theory of coupled diffusion and large deformation in polymeric gels. *J. Mech. Phys. Solids* 56, 1779-1793 (2008).

28. Gu, M., Yang, H., Perea, D. E., Zhang, J. G., Zhang, S. & Wang C. M. Bending-induced symmetry breaking of lithiation in germanium nanowires. *Nano Lett.* 14, 4622-4627 (2014).

5. The text on page 15 refers to Supplementary Table 1, but this table is missing in the supplementary information.

Response: Supplementary Table 1 has been addition to the supplementary information.

	Mass Diffusion	Thermal Diffusion
Governing equation	Kinetics: $\vec{j} = -D\vec{\nabla}C$ Equilibrium: $\frac{\partial C}{\partial t} = \vec{\nabla}(D\vec{\nabla}C)$ C is the concentration. D is the diffusion coefficient.	Kinetics: $\vec{q} = -k\vec{\nabla}T$ Equilibrium: $\frac{\partial T}{\partial t} = \vec{\nabla}\left(\frac{k}{\rho c}\vec{\nabla}T\right)$ k is the thermal conductivity, ρ is the density and c is the specific heat.
Volumetric strain	$\gamma_v \Delta C$ γ_v is the diffusion-induced expansion coefficient	$\alpha_v \Delta T$ α_v is the thermal expansion coefficient

Table R1. Comparison between mass-diffusion model and thermal-diffusion model